# Comparative transcriptome analysis provides new insights into the response of common bean to infection by race 65 of *Colletotrichum lindemuthianum*

Caroline Marcela da Silva Dambroz[1☯], Alexandre Hild Aono[2☯], Larissa Carvalho Costa[3☯], Evandro Novaes[1], Welison Andrade Pereira[1]*

1 Department of Biology, Federal University of Lavras, Lavras, Minas Gerais, Brazil, 2 Molecular Biology and Genetic Engineering Center (CBMEG), University of Campinas (UNICAMP), Campinas, SP, Brazil, 3 Plant Germplasm Quarantine Program, Animal and Plant Health Inspection Service, United States Department of Agriculture, Beltsville, Maryland, United States of America

☯ These authors contributed equally to this work.
* welison.pereira@ufla.br

## Abstract

The farming of common beans (*Phaseolus vulgaris*) is crucial for global nutrition, culture, and economy, but the crop faces significant challenges from biotic and abiotic stresses. Among these, anthracnose caused by *Colletotrichum lindemuthianum*, particularly race 65, is notable due to its widespread occurrence and high genetic and virulence diversity, especially in tropical regions such as Brazil. Understanding the molecular basis of resistance and susceptibility to specific pathogen races is key to accelerating the development of superior cultivars. Despite its significance, global gene expression studies addressing the interactions between bean genotypes and race 65 remain scarce. In this study, we explored the molecular basis of resistance and susceptibility to race 65 in two Brazilian common bean cultivars. RNA was extracted from leaves at 0, 48, and 96 hours after inoculation and sequenced using the Illumina NextSeq 500 platform. Our transcriptome analysis identified several candidate genes linked to resistance, including those involved in pathogen recognition, such as kinases and NB-LRR (nucleotide-binding and leucine-rich repeat) genes, as well as genes involved in the phenylpropanoid, glycerolipid biosynthesis, linoleic acid pathways, and cell wall remodeling. In contrast, the susceptible genotype exhibited activation of auxin signaling and sugar transport genes. Additionally, gene coexpression network analysis revealed a strong correlation among resistance-related genes. These findings provide valuable insights into the molecular-genetic mechanisms underlying common bean resistance to race 65 of *C. lindemuthianum*.

**Data availability statement:** We inform you that our data have been deposited in the NCBI and are publicly available under the accession number PRJNA1219552 (SRA).

**Funding:** This study was supported by scholarships and grants from Brazilian funding agencies: Conselho Nacional de Desenvolvimento Científico e Tecnológico (CNPq; 142203/2019-0 to Caroline Marcela da Silva Dambroz and 314269/2023-1 to Evandro Novaes), Fundação de Amparo à Pesquisa do Estado de São Paulo (FAPESP; 2019/03232-6 to Alexandre Hild Aono), and Fundação de Amparo à Pesquisa do Estado de Minas Gerais (FAPEMIG; APQ-00511-23 to Welison Andrade Pereira).

**Competing interests:** The authors have declared that no competing interests exist.

## Introduction

Anthracnose, caused by the hemibiotrophic fungus *Colletotrichum lindemuthianum* (Sacc & Magn) Scribn, is one of the most economically significant diseases affecting common beans [1]. This disease is particularly prevalent in tropical and subtropical regions where climatic conditions favor pathogen development [2]. It impacts the quality of grains and pods and can severely reduce crop yields. Integrated disease management strategies include cultural practices, chemical control, and the use of resistant cultivars, with genetic resistance being considered the most efficient and safe method [3]. However, developing common bean cultivars with durable resistance to this pathogen is challenging, mainly due to the high virulence diversity of *C. lindemuthianum* [2].

The resistance of common beans to *C. lindemuthianum* has been characterized as predominantly qualitative, following the gene-to-gene model [4]. Approximately 23 resistance loci with independent effects have been identified in different common bean varieties [5]. In most cases, resistance is conferring by the dominant allele; however, recessive resistance genes such as *co*-8 [6] and, more recently, *co-Indb* [7] have also been reported to confer anthracnose resistance in common bean. Despite the predominance of major resistance genes, quantitative trait loci (QTLs) conferring partial resistance to different races of *C. lindemuthianum* have also been identified [8–10]. Genetic mapping has revealed that these resistance loci are organized into complex clusters, comprising closely linked genes that confer resistance to specific races of the pathogen [3,9,11,12].

Race 65 is one of the most prevalent races of *C. lindemuthianum* in Brazil [13]. Although several major resistance loci (*Co*-01, *Co*-02, *Co*-03, *Co*-04, *Co*-05, *Co*-06 and *Co*-15) have been described as conferring resistance to race 65, recent studies demonstrate the specificity of resistance genes for different isolates within this race [14]. The high genetic and virulence diversity identified among isolates of race 65 [14–16] hinders the development of cultivars with durable genetic resistance and facilitates their widespread distribution in bean-growing regions. The combination of prevalence, aggressiveness, and genetic diversity of this race in Brazilian bean fields poses a persistent threat to bean production. Although knowledge of loci associated with common bean resistance to different races of *C. lindemuthianum* has advanced, a comprehensive understanding of the genes within these loci, their functional mechanisms, and interactions in host defense pathways remain crucial. This knowledge is essential for developing more effective strategies to obtain common bean cultivars with durable resistance to this pathogen [17].

Proteins encoded by resistance-related genes can act at various stages of the defense process, including pathogen recognition, triggering secondary defense responses, and participating in metabolic pathways [18]. For example, genes encoding proteins with kinase domains can function as pattern recognition receptors (PRRs). These receptors recognize pathogen-associated molecular patterns (PAMPs) and activate the defense response [19]. Within the superfamily of protein kinase-encoding genes (kinome), several leucine-rich repeat receptor-like protein kinases (LRR-RLK) have been associated with anthracnose resistance loci [19–21].

Other genes identified within resistance loci encode proteins with nucleotide-binding domains and leucine-rich repeats (NBS-LRR) [20]. These proteins can recognize pathogen effectors through protein-protein interactions, monitor the presence and function of other important host proteins, and induce effector-triggered immunity (ETI) upon detecting any alterations [22]. In all cases, the activation of the defense response initiates a signaling cascade, with proteins involved in different metabolic pathways triggering a series of responses that culminate in successful plant defense [18].

Several studies have investigated the differential expression of candidate genes in *Phaseolus vulgaris* cultivars resistant and susceptible to *C. lindemuthianum*, particularly race 73 [23–25], highlighting different perspectives on response mechanisms. Transcriptomic and gene expression analyses have highlighted key defense mechanisms, such as the role of genes associated with the *Co-1* [25], and *Co-4²* [23,24,26] loci, as well as the regulation of transcription factors and the modulation of hormonal pathways, including ethylene, cytokinin, jasmonic acid, and salicylic acid [23]. Similar findings have been reported for other races, such as 65 [26], 81 [27], and C531/100 [28], reinforcing the importance of PR proteins and the PAL biosynthetic pathway in resistance responses. Additionally, studies involving multiple pathotypes [29] have demonstrated that the timing and intensity of gene expression are key factors in determining compatible and incompatible interactions. In the case of race 89, different genes are modulated throughout the biotrophic and necrotrophic phases of infection [30].

Given the specificity of genetic control in response to different races of *C. lindemuthianum*, coupled with the importance of race 65 for the crop, our study aimed to explore the transcriptional profiles of two common bean cultivars, one resistant and one susceptible to race 65 across an infection time course following artificial inoculation. Our findings offer valuable insights into the molecular mechanisms involved in the response of *P. vulgaris* to this specific isolate of race 65, thereby enhancing our comprehension of the molecular-genetic mechanisms involved in both compatible and incompatible interactions between *P. vulgaris* and *C. lindemuthianum*.

## Materials and methods

### Plant material

The common bean cultivars BRS Estilo and Ouro Vermelho, both of Mesoamerican origin and commercially available in Brazil, were selected for this study. These cultivars showed segregation for a single major locus in response to *C. lindemuthianum* race 65 (Isolate Lv134): Ouro Vermelho carries dominant resistance alleles, while BRS Estilo is susceptible [14].

To confirm this contrast, seeds were sown in polyethylene trays with Plantmax® substrate and grown under greenhouse conditions (95% RH, 24°C). After primary leaf expansion, plants were inoculated with the fungal isolate to assess symptoms. The experiment followed a completely randomized 2×3 factorial design (cultivars×collection times) with three replicates per treatment.

### Preparation of the conidia suspension and inoculation with *Colletotrichum lindemuthianum* race 65

The *C. lindemuthianum* isolate Lv134 (race 65) is part of the fungal collection at the Universidade Federal de Lavras (Brazil). Mycelium fragments were transferred from M3 medium plates [31] to sterilized bean pods in test tubes containing solidified water agar [32] and incubated in the dark at 22°C for 15 days to enhance sporulation.

For inoculum preparation, conidia were scraped from pod surfaces, filtered through gauze, and adjusted to $1.2 \times 10^6$ conidia/ml using a Neubauer chamber [33]. The suspension was sprayed on both sides of leaves and stems until runoff [34]. After inoculation, plants were kept in a mist chamber (95% RH, 25±2°C) for 48h, then transferred to a greenhouse (85% RH, 24°C) for seven days before symptom assessment. Control plants were inoculated with water.

### Collection of leaf material and assessment of symptoms

Leaf samples were collected at 0, 48, and 96 hours after inoculation (hai). Half of each primary leaf was flash-frozen in liquid nitrogen and stored at −80°C for RNA extraction, while the other half remained on the plant for symptom assessment.

Nine days post-inoculation, plants were evaluated using a 1–9 scale [35], with plants scoring ≤3 classified as resistant and those scoring >3 as susceptible. Statistical analysis was performed using Student's *t*-test ($p \leq 0.05$) in R [36].

## RNA extraction, cDNA library construction, and sequencing

Total RNA was extracted from 12 samples (BRS Estilo and Ouro Vermelho, three time points: 0, 48, and 96 hai, two replicates each) using RNATRIzol® (Invitrogen) with modifications [37]. Residual DNA was removed with TURBOTM DNAse (ThermoFisher). RNA concentration and integrity were assessed via Qubit (Qiagen) and Agilent 4200 TapeStation. The mRNA was enriched using NEBNext® Poly(A) Magnetic Isolation (New England BioLabs), and libraries were prepared with the TruSeq Stranded Total RNA Library Plant Kit (Illumina). Libraries were quantified, normalized, and sequenced (2×75 bp, paired-end) on the Illumina NextSeq 500.

## Bioinformatics analysis

Sequencing quality was assessed with FastQC v0.11.7 [38]. Adapter sequences and bases with Q<15 (four-base window) were trimmed using Trimmomatic v0.39 [39], discarding reads<32 bases. Filtered reads were aligned to the *P. vulgaris* v2.1 genome (Phytozome v.13) [40] using STAR v2.7.10b [41] with predefined parameters, including mismatch filtering and gene quantification: --outFilterType BySJout --outFilterMultimapNmax 20 --alignSJoverhangMin 8 --alignSJDBoverhangMin 1 --outFilterMismatchNmax 999 --outFilterMismatchNoverLmax 0.10 --outFilterMatchNmin 50 --outFilterScoreMinOverLread 0 --outFilterMatchNminOverLread 0 --alignMatesGapMax 1000 --alignIntronMax 3000 --quantMode GeneCounts.

## Differential expression analysis and enrichment of Gene Ontology terms

Differential expression analysis was performed using DESeq2 v1.36.0 [42] in R v4.2.1 [38] on STAR-mapped read counts. Genes with <10 total reads were filtered out. Hierarchical clustering (UPGMA, Spearman correlations) and PCA were also performed. Differentially expressed genes (DEGs) were identified for six contrasts (Table 1), comparing cultivars at each time point and time points within cultivars. Significance was set at FDR-adjusted $p < 0.05$ and log2 fold-change ≥ 2.0 (TPM-based).

Genes upregulated in contrasts 1 and 2 or upregulated in contrast 1 and downregulated in contrast 3 were classified as putative R48-genes, while the opposite pattern defined putative S48-genes. The same logic applied to contrasts 4–6 (Table 1). Further analysis identified overlaps between putative R-genes and S-genes, resulting in resistance-like (R48-like) and susceptibility-like (S48-like) genes at 48 hai. This methodology was extended to identify R96-like and S96-like genes. Expression profiles, log2-normalized across treatments, were visualized using a heatmap generated from UPGMA clustering in pheatmap v.1.0.12 [43].

**Table 1. Contrasts analyzed for differential gene expression considering the resistant (Ouro Vermelho) and susceptible (BRS Estilo) common bean cultivars to Lv134 of race 65 of *Colletotrichum lindemuthianum* in different collection times (0, 48, and 96 hai).**

| Contrast | | Up-regulated genes | Down-regulated genes |
|---|---|---|---|
| 1 | OV 48hai x E 48hai | Putative R48-genes[a] | Putative S48-genes[b] |
| 2 | OV 48hai x OV 0hai | Putative R48-genes[a] | Putative S48-genes[b] |
| 3 | E 48hai x E 0hai | Putative S48-genes[b] | Putative R48-genes[a] |
| 4 | OV 96hai x E 96hai | Putative R96-genes[c] | Putative S96-genes[d] |
| 5 | OV 96hai x OV 48hai | Putative R96-genes[c] | Putative S96-genes[d] |
| 6 | E 96hai x E 48hai | Putative S96-genes[d] | Putative R96-genes[c] |

[a]R48-like genes: Intersection between Putative R48-genes; [b]S48-like genes: Intersection between Putative S48-genes

[c]R96-like genes: Intersection between Putative R96-genes; [d]S96-like genes: Intersection between Putative S96-genes

## Gene functional profiling

Gene Ontology (GO) term enrichment analyses of the DEGs were performed using the topGO R v.2.48.0 package [44], applying Fisher's exact tests with an FDR-adjusted p-value cut-off of 0.05. Based on Phytozome v.13 annotations [45], selected DEGs were analyzed for involvement in metabolic pathways. Enriched pathways among gene groups from intersecting contrasts were also explored using Enzyme Consortium (EC) numbers and the KEGG database [46], with Fisher's exact tests (p-value of 0.05) in R.

## Gene coexpression networks

Coexpression networks were constructed using Pearson correlations (minimum correlation coefficient of 0.9) and the HRR method [47], with a threshold of 30 correlations for significant connections. Two networks were built with the R package igraph v.1.3.5 [48] for the susceptible (BRS Estilo) and resistant (Ouro Vermelho) genotypes. Nodes related to DEG intersection groups (S-like and R-like genes) were selected, and the initial gene selection was extended to include first-degree neighbors. Hub scores were calculated using Kleinberg's hub centrality scores [49].

## qPCR validation of differentially expressed genes

Ten DEGs and the actin gene (*Phvul.011G064500.1*) were selected for qPCR validation (S1 Table). Primers and probes were designed using Primer3Plus with specified parameters (product size: 80–120 bp, primer size: 18–22 nt, GC content: 40–60%, 1.5 mM divalent cations, and 0.6 mM dNTPs) to avoid primer dimers and hairpins. cDNA synthesis used 1 µg RNA, random hexamers, and a dNTP mix. Reverse transcription was performed with Maxima H Minus Enzyme Mix. qPCR reactions were run on a QuantStudio 3 system using PrimeTime Gene Expression Master Mix (Integrated DNA Technologies, Coralville, IA, USA), with two technical replicates for each biological replicate. The mean Cq was corrected using the actin gene, and Pearson correlations between RNA-Seq expression and deltaCq values were performed in R v.4.2.1. Log2 fold change differences across contrasts were visualized in bar plots.

## Results

### Reaction of the common bean lines to isolate Lv134 of race 65 of *Colletotrichum lindemuthianum*

The first anthracnose symptoms appeared at 96 hai on BRS Estilo plants (average disease score of 8.78), while Ouro Vermelho plants remained symptom-free (average score of 1.27) throughout the trial (Fig 1A-C). In susceptible plants, lesions worsened, resulting in plant death within seven days. These results align with the expected compatible and incompatible interactions between the pathogen and the susceptible and resistant genotypes, respectively.

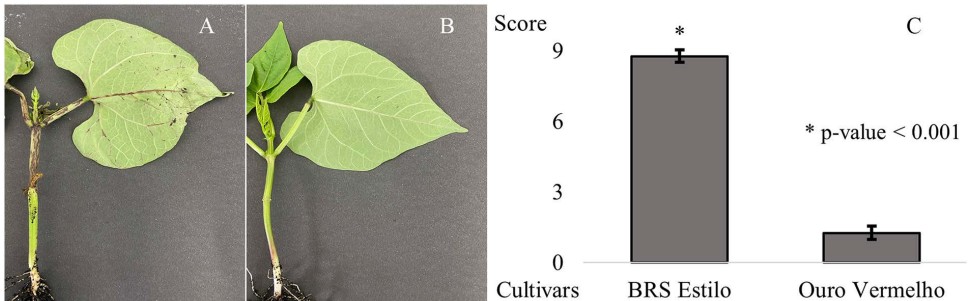

**Fig 1. Phenotypic evaluation of BRS Estilo (A) and Ouro Vermelho (B) cultivars, seven days after inoculation with *Colletotrichum lindemuthianum* isolate Lv 134, race 65. Student's t-test applied to the mean anthracnose severity scores; * indicates significance at 0.1% probability.**

## RNA-Seq results

After filtering with Trimmomatic, 12 RNA-Seq libraries retained 17–35 million reads (average 30 million). Mapping to the *P. vulgaris* v2.1 genome showed 90% unique reads. PCA indicated clustering of BRS Estilo and Ouro Vermelho replicates by collection time (Fig 2A), explaining 79% of variation. The dendrogram confirmed replicate and cultivar clustering by time point (Fig 2B). qPCR validation of 10 differentially expressed genes at 0, 48, and 96 hours after inoculation (S1 Fig) corroborated RNA-Seq results, confirming data quality for further analysis.

## Differentially expressed genes over the time course of infection

We evaluated the expression of 22,344 genes over the infection period (S3 Table) using 6 contrasts, identifying DEGs at 48 or 96 hai for each genotype (Figs 3A and 3F). Initially, all DEGs were considered potential candidates for resistance or susceptibility, but this list was refined based on their expression across the contrasts [17].

In contrast 1 (OV 48 hai x E 48 hai), 770 DEGs were identified, with 353 downregulated and 417 upregulated in the resistant cultivar. In contrast 2 (OV 48 hai x OV 0 hai), 6,474 DEGs were found (3,742 downregulated, 2,732 upregulated), and in contrast 3 (E 48 hai x E 0 hai), 4,855 DEGs were identified (2,761 downregulated, 2,094 upregulated) (Fig 3A). Similar contrasts were done at 96 hai (Fig 3F).

Specific intersections at 48 hai showed 88 genes upregulated in contrasts 1 and 2 (Fig 3B), and 108 upregulated in contrast 1 and downregulated in contrast 3 (Fig 3C), identified as putative R48-genes (S4 Table). Conversely, 101 genes were downregulated in both contrasts 1 and 2 (Fig 3D) and 101 genes downregulated in contrast 1 and upregulated in contrast 3 (Fig 3E), as putative S48-genes (S5 Table).

At 96 hai, 620 genes were upregulated in contrasts 4 and 5 (Fig 3G), while 105 were upregulated in contrast 4 and downregulated in contrast 6 (Fig 3H), identified as putative R96-genes (S6 Table). Conversely, 174 genes were downregulated in both contrasts 4 and 5 (Fig 3I) and 193 were downregulated in contrast 4 and upregulated in contrast 6 (Fig 3J), as putative S96-genes (S7 Table). This approach further refines the repertoire of DEGs potentially associated with the pathosystem across different time points.

Samples were clustered based on normalized expression values of putative R and S genes (Fig 3). Several genes that were unexpressed or downregulated at 48 hai in Ouro Vermelho were upregulated at 96 hai, indicating major

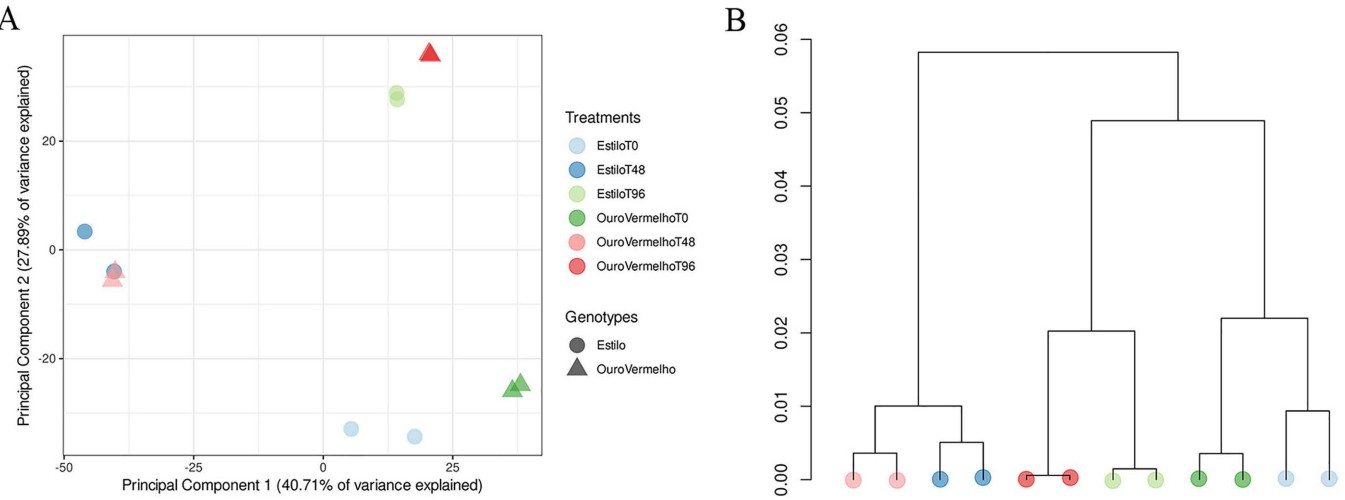

**Fig 2. PCA showing similarity in gene expression patterns of common bean in response to *Colletotrichum lindemuthianum* (A). UPGMA dendrogram of gene expression levels across common bean samples (B).**

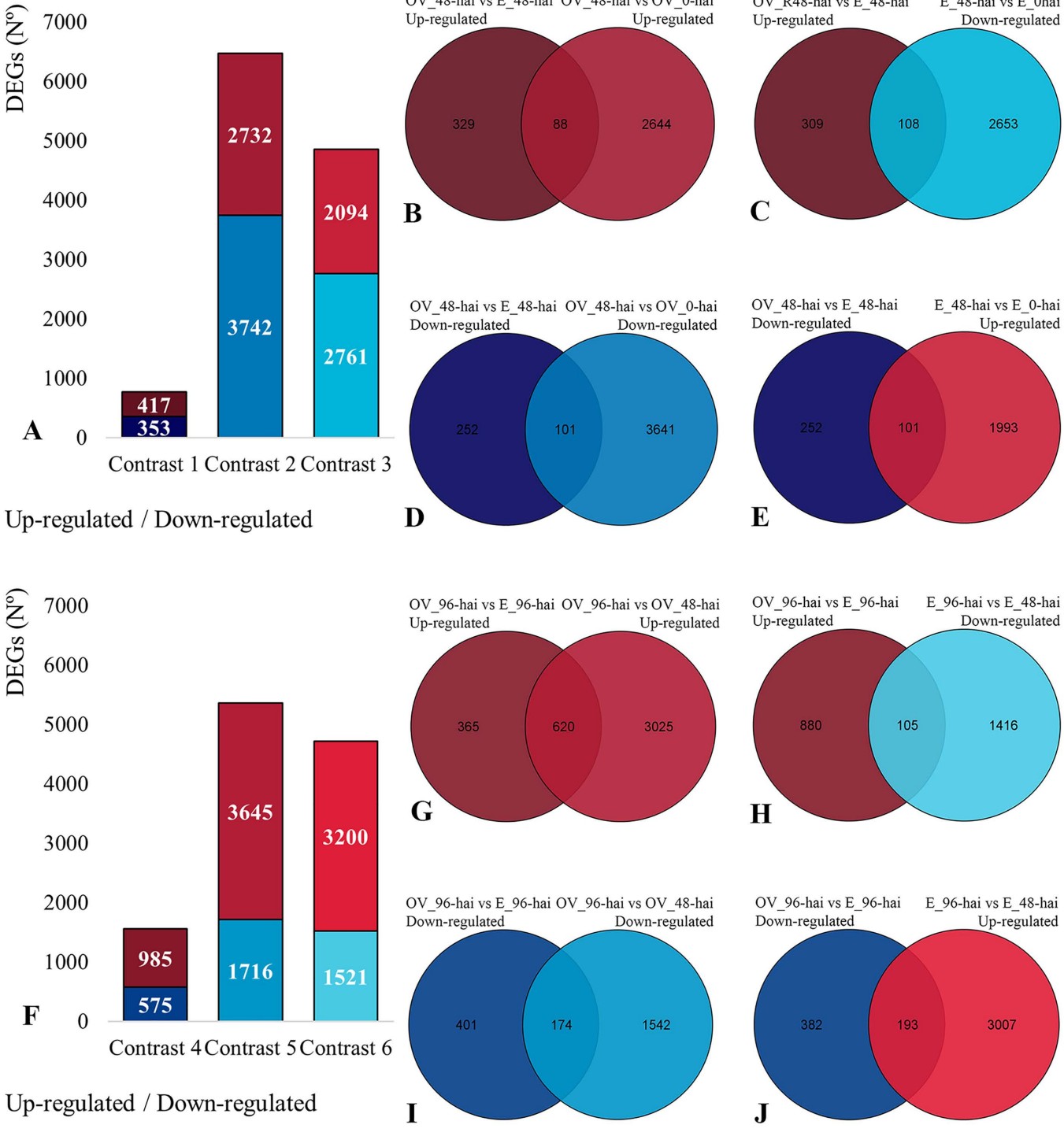

**Fig 3. Differentially expressed genes in contrasts between cultivars and inoculation times, highlighting intersections for resistance and susceptibility associations.** Number of upregulated and downregulated DEGs in contrasts 1, 2, and 3 **(A)**. Venn diagrams for putative R48-genes **(B, C)**, and for putative S48-genes **(D, E)**. DEGs in contrasts 4, 5, and 6 **(F)**. Venn diagrams for R96-genes **(G, H)**, and for S96-genes **(I, J)**.

transcriptional reprograming during infection (Fig. 4). A similar pattern was observed in BRS Estilo, where genes upregulated at 48 hai were repressed at 96 hai, and genes downregulated at 48 hai were induced at 96 hai. These contrasting dynamics highlight the progression of host responses over time, with putative genes frequently shifting from negative to positive regulation, or vice versa, by 96 hai (Fig 4).

To identify a core set of candidate genes involved in the plant defense response against the pathogen, we employed a more stringent approach. Genes were considered to be associated with resistance if they were significantly up-regulated in contrasts 1 and 2. To provide even stronger evidence for their role in resistance, these candidates genes should also be significantly downregulated in contrast 3, i.e., in the susceptible cultivar (S8 Table; Fig 5A-D). At 48 hai, three genes met these criteria and were designated as resistance-like genes (R48-like genes). Conversely, eight genes at 48 hai were found to be negatively regulated in contrasts 1 and 2, as well as positively regulated in contrast 3, and were therefore classified as susceptibility-like genes (S48-like genes). Applying the same criteria at 96 hai, we identified nine R96-like genes and seven S96-like genes.

## Functional profiling

GO enrichment analysis revealed distinct functional roles for DEGs in the putative R- and S-genes groups. DEGs considered as putative R48 genes and putative S48 genes were enriched for six and seven GO terms of biological processes,

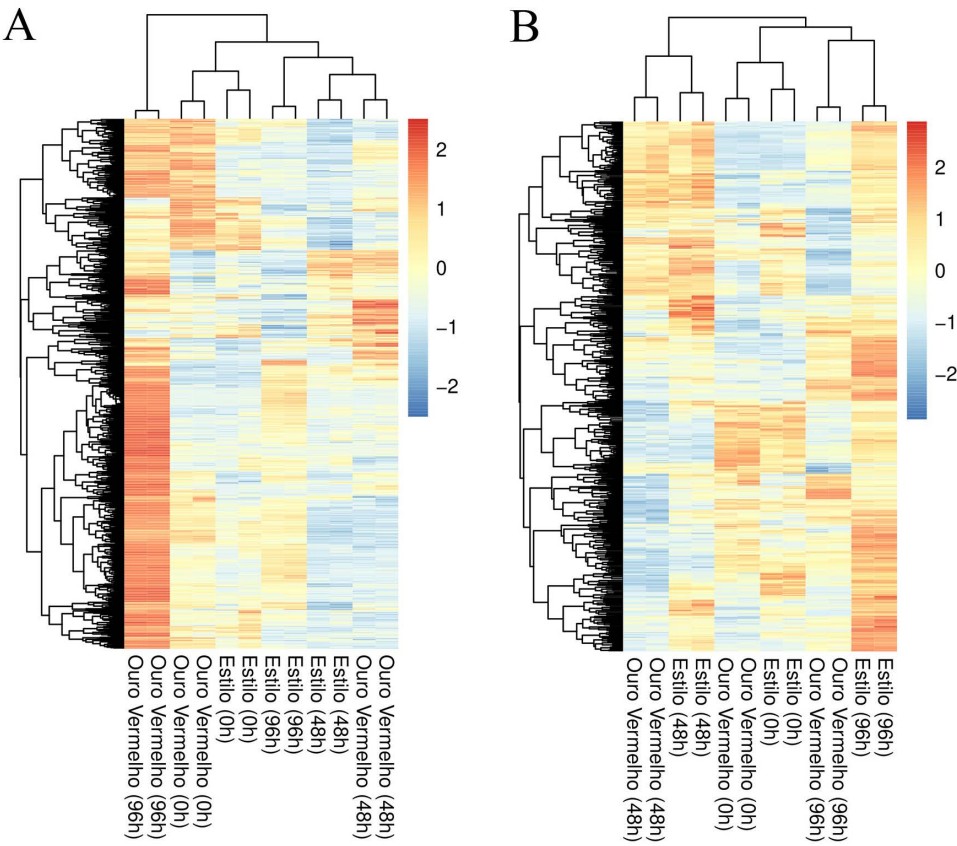

**Fig 4. Hierarchical cluster analysis of normalized gene expression patterns in *Ouro Vermelho* and *BRS Estilo*.** The heatmap displays the expression levels of significant differentially expressed genes within the putative R-genes (A) and putative S-genes (B) groups in plants inoculated with isolate Lv 134 of race 65 of *Colletotrichum lindemuthianum* at 0, 48, and 96 hours after inoculation (hai). Genes that are positively regulated are shown in red, while negatively regulated genes are represented in blue.

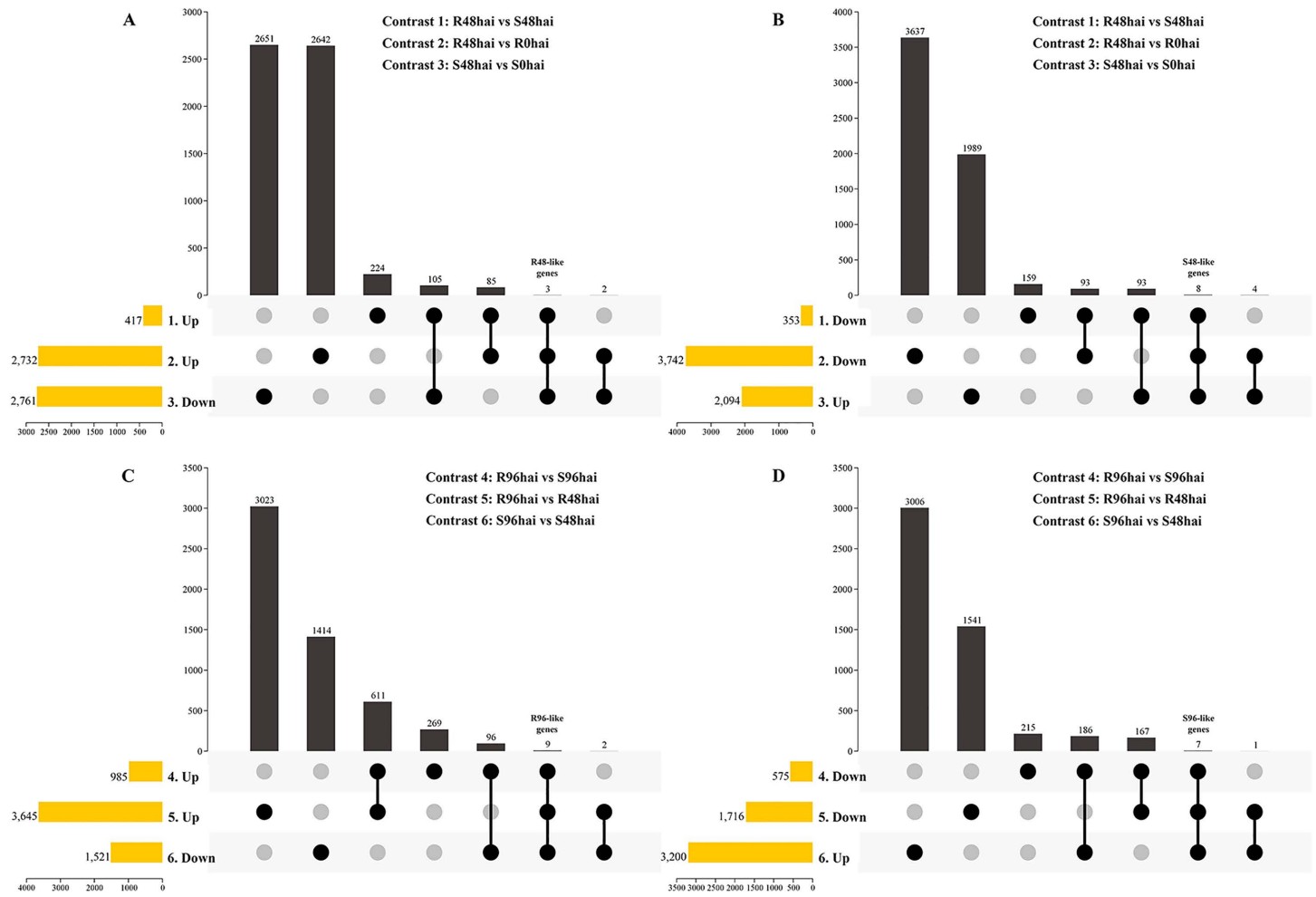

**Fig 5. Upset plots used to highlight shared genes between three contrasts and specific conditions to indicate R48-like genes (A), S48-like genes (B), R96-like genes (C), and S96-like genes (D).**

respectively (S9 Table). The enrichment of "response to biotic stimulus" (GO:0009607) in both genotypes, is consistent with a shared, general plant response to the presence of the pathogen, as also observed by [28]. Following this, the significant representation of the "metabolic process" (GO:0008152) suggests that transcriptional reprogramming of plants metabolism is also an important component of the resistance response. Additionally, the enrichment of "photosynthesis, light harvesting" (GO:0009765) and the "cellulose biosynthetic process" (GO:0030244) reflect specific defense strategies that involve the reallocation of energy resources and the reinforcement of the cell wall, respectively. Among the putative S48-genes, in addition to the "response to biotic stimulus" (GO:0009607), enrichments of the "defense response" (GO:0006952) and "response to oxidative stress" (GO:0006979) were observed, which may reflect the stress brought by the pathogen to the plant. Furthermore, the dynamics of the plant's response to stress include the "protein insertion into mitochondrial outer membrane" (GO:0045040), the "cellular glucan metabolic process" (GO:0006073), the "cell wall modification" (GO:0042545), and "protein ubiquitination" (GO:0016567). As pathogen infection progressed to 96 hai, the "cellulose biosynthesis process" (GO: 0030244), which was also enriched in the set of putative R48 genes, was the only process that remained among the putative R96 genes. Among the putative S96-genes, the "cell wall modification"

(GO:0042545) continued to be enriched, a process that was also prominent in the S48-genes. Among the putative S96-genes, "cell wall modification" (GO:0042545) continued to be enriched, a process that was also prominent in the S48 genes. The "cell wall macromolecule catabolic process" (GO:0016998) was particularly enriched among S96-genes These results confirm distinct transcriptional dynamics between plants that are susceptible and those that are resistant to the pathogen.

We identified enriched KEGG pathways in the putative R-genes and S-genes groups (S10 Table). In the putative R-genes group (p-value ≤ 0.01), enriched pathways included phenylpropanoid biosynthesis, involving arogenate dehy-dratase genes *Phvul.006G149600* and *Phvul.005G127100*, linoleic acid metabolism, evidenced by lipoxygenase genes (*Phvul.006G185300*, *Phvul.005G156800*, *Phvul.009G262900*), and glycerolipid metabolism, represented by glycerol-3-phosphate acyltransferase genes (e.g., *Phvul.002G192000*). The putative S-genes group (p-value ≤ 0.01) also showed enrichment in phenylpropanoid biosynthesis, along with flavonoid, tropane, piperidine, and pyridine alkaloid, sesquiter-penoid, and triterpenoid biosynthesis pathways.

### Gene coexpression networks

We constructed four coexpression networks to explore gene association patterns: (i) Ouro Vermelho with putative R-genes (S11 Table); (ii) BRS Estilo with putative R-genes (S12 Table); (iii) Ouro Vermelho with putative S-genes (S13 Table); and (iv) BRS Estilo with putative S-genes (S14 Table).

In the resistant genotype, the R-gene network exhibited the highest interconnectivity (S11 Table; Fig 6A), with hub genes such as *Phvul.002G106300* (GDSL-like Lipase/Acylhydrolase), *Phvul.002G076400* (matrix metalloproteinase), and *Phvul.002G107900* (pathogenesis-related protein) showing high connectivity (130, 116, and 112 connections, respectively) and differential expression at 96 hai. Additionally, genes involved in pathogen recognition, including protein kinases (e.g., *Phvul.002G115800*, *Phvul.002G115900*, *Phvul.003G024000*) and NBS-LRR proteins (*Phvul.002G021700*), were well-represented, particularly on chromosomes 2 and 3. Other DEGs, such as *Phvul.009G080000* (WRKY) and *Phvul.005G124100* (bZIP), were also part of the R-gene network 48 hai, alongside genes involved in secondary carbohy-drate metabolism (e.g., *Phvul.005G082800*, beta glucosidase, 96 hai).

Within the coexpression network of putative R-genes in the susceptible genotype (S12 Table), the genes *Phvul.002G106300* and *Phvul.002G022600*, encoding GDSL-like Lipase/Acylhydrolase proteins, had high connection values (82 and 59 degrees), while *Phvul.002G152700*, encoding leucoanthocyanidin dioxygenase linked to flavonoid bio-synthesis, had 78 connections. [50]

In the putative S-genes network of the resistant genotype, *Phvul.002G096500* (UDP-glucosyl transferase) was central with 102 connections (S13 Table). In the susceptible genotype, putative S-genes network, *Phvul.002G073100* (MtN21/EamA-like nodulin, 71 connections) and *Phvul.002G154600* (MATE efflux protein, 66 connections) were more expressed at 48 hai. The gene with the most connections in the susceptible network was *Phvul.002G096500* (125 connections), also upregulated in the susceptible genotype (S14 Table).

### Differentially expressed genes located near previously identified loci associated with resistance to race 65 of *Colletotrichum lindemuthianum*

Previous studies have linked the common bean loci *Co*-01, *Co*-02, *Co*-03, *Co*-04, *Co*-05, *Co*-06, and *Co*-15 to resistance against *C. lindemuthianum* race 65 [5,14,20,50–52]. In our study, we identified DEGs near these loci (500 Kb window) within the coexpression network of the resistant genotype. For *Co*-01, preciously mapped on Pv01 (49,583,965 bp) using the ss715645251 marker [11], 13 DEGs were detected. Of these, only *Phvul.001G243300* (C2 phospholipase) was included in the R-gene network, and it was exclusively and differentially expressed in the resistant genotype at 96 hai.

For *Co*-04 locus, located on Pv08 (SAS13 marker, 2,281,755 bp) [53], four DEGs encoding RLK-LRR proteins with malectin domains, was found into the R-gene network, including *Phvul.008G028400*, which was upregulated

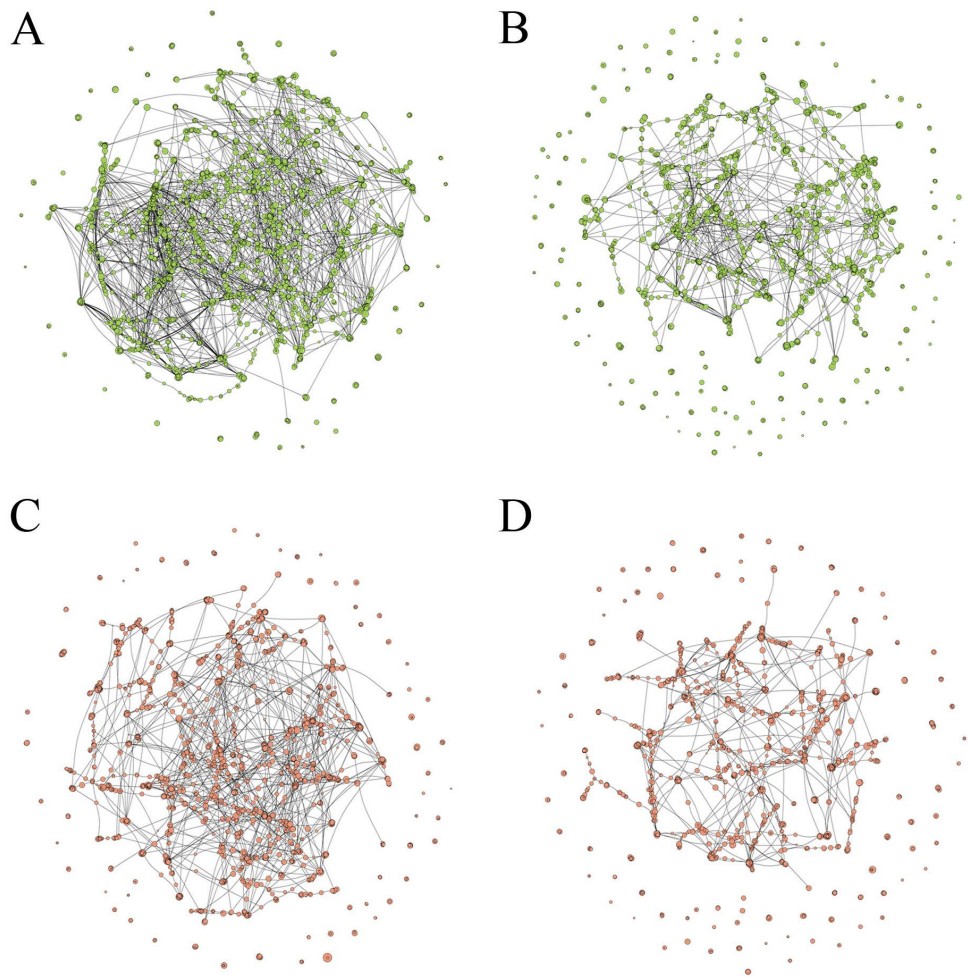

**Fig 6. Co-expression networks of putative R-genes and S-genes identified in contrasts at 0, 48, and, 96 hours post-inoculation of Ouro Vermelho (resistant) and BRS Estilo (susceptible) cultivars with *Colletotrichum lindemuthianum* race 65, isolate Lv 134.** Putative R-genes from the resistant genotype **(A)**; Putative R-genes from the susceptible genotype **(B)**; Putative S-genes from the resistant genotype **(C)**; Putative S-genes from the susceptible genotype **(D)**.

at 96 hai in the resistant genotype and integrated into the R-gene network. For *Co*-06 on Pv07 (SZ04 marker) [54], *Phvul.007G095300* (GATA transcription factor) was positively regulated in the resistant genotype at 96 hai.

## Discussion

The ongoing threat that *C. lindemuthianum* poses to global common bean production underscores the need for effective disease management strategies. Our study provides new insights into the molecular basis of the compatible and incompatible interactions between common bean cultivars and race 65 of this pathogen. Race 65 is a particularly important target for common bean breeding programs, especially in Brazil, due to its high genetic diversity and the frequent emergence of new virulent isolates [5,15,16,55–57]. By dissecting the host's molecular response, our research offers a foundational understanding that can guide the development of more durable resistance to this pathogen.

The time selected for this stydy (48 and 96 hai) corresponds to key infection stages of the pathogen and allowed the analysis of the dynamic nature of plant-pathogen interactions [26]. Following inoculation, conidia of *C. lindemuthianum*

adhere to the plant surface, forming appressoria for penetration. This is followed by infection vesicle formation and primary hyphae differentiation, marking the biotrophic phase, which lasts for approximately 72 hours, with the host cell remaining alive [58]. The fungus then transitions to the necrotrophic phase, characterized by secondary hyphae and host cell death [25,26,59]. Macroscopic symptoms appear at 96 hai [58,60], which was confirmed in the susceptible cultivar (BRS Estilo) in this study.

To assess gene expression alterations in this pathosystem, we analyzed transcriptomes from resistant and susceptible plants to *C. lindemuthianum* race 65, using six contrasts (Table 1). This approach revealed marked transcriptional variation, and the intersection of contrasts enabled the discrimination of DEGs with potential roles in resistance or susceptibility. Genes consistently regulated across contrasts were classified as putative R48 or S48 candidates (Fig 3; S4 Table), representing responses more directly associated with the host phenotype rather than general effects of pathogen challenge. This systematic strategy provided a robust framework to refine the set of candidate genes for subsequent functional interpretation.

Among the putative R48 genes, we identified alpha-beta hydrolases, disease resistance proteins with NB-ARC domains, and proteins with kinase domains, consistent with findings in other *C. lindemuthianum – P. vulgaris* pathosystems [23–25,28]. The presence of NB-ARC domains and kinase proteins in the resistant cultivar highlights the activation of critical components of the plant's immune system, specifically involved in pathogen recognition and signal transduction to initiate downstream defense responses [21]. Conversely, the set of S48 genes include disease resistance proteins, TIR-NBS-LRRs, RLK-LRRs, responsible for transducing signals among other functions [21], concanavalin A-type lectin family protein kinases, and integrase-type DNA-binding proteins. At 96 hai, additional candidates emerged, such as protein kinases, bHLH, and sulfotransferase 2A among R96 genes, while the S96 genes include disease resistance proteins, RING/U-box proteins, and RLK-LRRs. Previous studies have identified LRR-RLK genes that are positively regulated in both resistant and susceptible lines to anthracnose [25]. The recurrent detection of immune receptors, kinases, and regulatory proteins across both time points emphasizes that the plant response to infection is highly dynamic, with certain gene classes consistently associated either with effective resistance or with susceptibility [28].

To refine the identification of candidate genes, we focused on those specifically regulated in individual contrasts, which revealed smaller sets of R48-, S48-, R96-, and S96-like genes (Fig 5A-D). Among the R48-like genes, hydrolases (*Phvul.002G332300*, *Phvul.L004500*) and a cytochrome P450 (*Phvul.008G277466*) were particularly noteworthy, as both are associated with lignin biosynthesis and cell wall modification [61,62]. P450 proteins are involved in lignin intermediate biosynthesis and other secondary metabolites, playing key roles in organism defense against pathogens [63]. The early upregulation of these genes suggests a rapid and robust activation of cell wall reinforcement mechanisms and the production of defensive secondary metabolites, crucial for containing fungal spread. In contrast, S48-like genes included candidates related to cellulose synthesis (*Phvul.005G001000*) [64], protease inhibition (*Phvul.004G129600*) [65], and secretion pathways (*Phvul.009G248400*) [64–66]. The presence of these genes in susceptible genotypes, particularly those related to general cellular processes or potential pathogen facilitation (e.g., protease inhibitors that might counteract plant defense proteases), indicates a different host strategy or a manipulation by the pathogen that leads to ineffective defense [63–65].

Two transcription factors from the PLATZ and WRKY families, *Phvul.002G061300* and *Phvul.005G181800*, identified among the R96-like genes [25,67] and their sustained upregulation in the resistant genotype at 96 hai suggests a central role in coordinating long-term antifungal defenses through transcriptional regulation. In contrast, S96-like genes included candidates involved in lignin biosynthesis, *Phvul.009G066300*, *Phvul.002G144600*, and *Phvul.001G145600* [68–70]. These findings suggest that both compatible and incompatible interactions induce a response at the cell wall level, but with differing outcomes. While lignin biosynthesis in resistant plants contributes to the formation of robust physical barriers, that restrict pathogen spread, its activation in susceptible plants may represent a delayed, insufficient response, or misdirected defense response [61,62].

Pathogen recognition in plants is largely mediated by pattern recognition receptors (PRRs), among which protein kinases constitute the major group. Genes like *Phvul.003G024000*, *Phvul.002G115800*, and *Phvul.002G115900* were upregulated at 96 hai, underscoring the relevance of protein phosphorylation in sustaining the defense response [20,21,71]. The sustained and amplified expression of these genes suggests a continuous signaling cascade in the resistant plant, which is crucial for maintaining an active defense state against the invading pathogen.

In addition to PRRs, pathogen recognition also can occur via intracellular NBS-LRR proteins. Some of these proteins were detected in both genotypes with higher activation in the resistant genotype (S3 Table). Their prominence in the coexpression network of putative R-genes is consistent with previous reports in the common bean-*C. lindemuthianum* pathosystem [25,28]. Their stronger induction of these genes in the resistant genotype points to a more effective perception of pathogen effectors, leading to a more robust effector-triggered immunity (ETI) compared to the susceptible genotype [28].

Upon pathogen recognition, a systemic defense response is triggered, which often involves salicylic acid (SA) and phytoalexins from the phenylpropanoid pathway [72]. In this study, genes encoding arogenate dehydratase (*Phvul.006G149600*, *Phvul.005G127100*), a precursor enzyme in this pathway, were enriched in the resistant genotype, reinforcing the role of this mechanism in defense, as also observed in *Glycine max*. Genes encoding arogenate dehydratase (*Phvul.006G149600*, *Phvul.005G127100*), which catalyzes a precursor step in this pathway, were enriched in the resistant genotype, reinforcing the potential role of phenylpropanoid pathway in plant defense, at mechanism also previously observed in *Glycine max* [73] and previous studies on phenylalanine ammonia-lyase (PAL2) expression in common bean infected with different *C. lindemuthianum* races [23,24,26,29]. The enrichment of the phenylpropanoid pathway in the resistant genotype indicates a rapid and potent activation of secondary metabolite synthesis [72], leading to the accumulation of antimicrobial compounds and precursors for cell wall strengthening, which are critical for effective defense against *C. lindemuthianum*.

Different lipoxygenase-encoding genes (*Phvul.006G185300*, *Phvul.005G156800*, *Phvul.009G262900*) from jasmonic acid (JA) pathway were differentially expressed in the resistant genotype, with *Phvul.009G262900* being activated at both 48 and 96 hai. Lipoxygenase activity has been associated with responses to pathogens in various crops, including tobacco and rice [74,75]. Additionally, a highly connected GDSL-like Lipase/Acylhydrolase (*Phvul.002G106300*), in the putative R-genes network, has a homolog in *Arabidopsis* that enhances resistance to *Sclerotinia sclerotiorum* via both SA- and JA-related pathways [76]. The concurrent activation of both SA and JA pathways in the resistant genotype suggests a finely tuned crosstalk between these hormone-mediated defenses [73], enabling a broad-spectrum response against the pathogen. The consistent activation of *Phvul.009G262900* indicates its sustained role throughout the infection process.

In the resistant genotype, glycerolipid metabolism also emerged as a potential resistance mechanism. Glycerol-3-phosphate and oleic acid regulate fatty acid metabolism and phytohormone crosstalk, inducing pathogenesis-related genes, as seen in wheat against powdery mildew [77]. This suggests that the resistant genotype actively remodels its lipid composition, potentially impacting membrane integrity, signaling molecule production, or even directly contributing to antimicrobial activity, thus serving as a novel component of the defense arsenal.

Our results also suggest a role for peroxisomal retrograde signaling. We observed a significant enrichment of genes related to peroxisome fission at 48 hai in the resistant genotype (S9 Table). As peroxisomes are central do producing reactive oxygen species (ROS) and defense-related phytohormones, including indole-3-acetic acid, JA, and SA, their enhanced activity points to a robust defense response [78]. Additionally, a matrix metalloproteinase, the second most connected gene in the R-gene network, may regulate ROS accumulation, a mechanism observed in other plant-pathogen interactions, such as in tomato against *Botrytis cinerea* [79].

Transcription factors (TFs) are central regulators of defense responses to biotic stress [80–82]. In the resistant genotype, WRKY and bZIP TFs were upregulated at 48 hai and integrated into the coexpression network of putative R-genes. Notably, WRKY (*Phvul.005G181800*) and PLATZ (*Phvul.002G061300*) were also identified among R96-like candidates, showing positive regulation in contrasts 4 and 5 and negative regulation in contrast 6. WRKY factors, in particular, are

linked to SA- and ethylene-mediated signaling, and have been implicated in plant responses not to only *C. lindemuthianum* in common beans [28], but also to other pathogens such as *Botrytis cinerea* and *Aspergillus flavus* in *Arabidopsis thaliana* and *Zea mays*, respectively [80,81]. In agreement, Padder et al. (2016) reported high TF expression in common bean after infection with race 73 of *C. lindemuthianum*, further supporting the view that transcriptional regulators represent a critical layer of the host response, though their activation may differ between resistant and susceptible genotypes.

Cell signaling often culminates in the synthesis of pathogenesis-related (PR) proteins, which display antimicrobial or enzymatic activities [83]. In our study, three major latex-like (MLP) genes (*Phvul.011G183500*, *Phvul.011G183766*, and *Phvul.011G183832*), members of the PR family, were identified within the network of putative R-genes of the resistant genotype. As subfamily of the Bet v_1 family, MLP proteins are involved in biotic and abiotic stress responses, contributing to disease resistance and stress tolerance [84]. MLP 28 enhances resistance to *Potato virus Y* in *Nicotiana benthamiana* [85]. Furthermore, the thaumatin-like PR gene (*Phvul.002G107900*) was strongly associated with other genes in the putative R-gene network and has been linked to wheat resistance to *Puccinia triticina* [86]. These findings highlight the integration of multiple PR proteins into the defense framework, underscoring their importance as conserved mediators of resistance.

Upon pathogen infiltration, plants reinforce the compromised cell wall by synthesizing new carbohydrates [87]. In this study, cellulose-related genes (*Phvul.002G136300*; *Phvul.005G001000*) were upregulated in the resistant genotype at 96 hai. Glycosyltransferase activity, essential for cell wall synthesis [64], persisted over time in R-genes, indicating a sustained response. Susceptible plants often exhibit deficiencies in papilla development and secondary wall thickening, compromising resistance [88]. Beyond cellulose, the resistant genotype showed increased expression of xyloglucan endo-transglucosylase/hydrolase genes (*Phvul.003G147300*, *Phvul.003G231900*), suggesting enhanced glucan deposition. Genes linked to pectin modification (*Phvul.005G011900*, *Phvul.005G184400*, *Phvul.009G222100*, *Phvul.010G123132*) were also upregulated, potentially strengthening cell wall integrity. Conversely, genes involved in carbohydrate metabolism, such as beta-glucosidases and hydrolases (*Phvul.005G082800*, *Phvul.005G130900*), were downregulated in resistant plants, suggesting a shift from primary metabolism and photosynthesis towards defense [89]. This comprehensive upregulation of genes involved in cell wall modification in the resistant genotype reveals a strategic reallocation of resources from growth to defense, actively building physical barriers to limit pathogen spread [86]. This contrasts sharply with the susceptible response, where such mechanisms are either absent, delayed, or insufficient, leading to successful colonization by *C. lindemuthianum*.

In the susceptible genotype, auxin-responsive SAUR proteins (*Phvul.006G022900*, *Phvul.009G001400*) were identified within the coexpression network. Auxin signaling has been repeatedly associated with disease promotion in diverse plant–pathogen interactions, including rusts, bacterial wilts, and blights [90–95]. Additionally, genes encoding sugar transporter (*Phvul.002G073100*, *Phvul.001G129500*, *Phvul.007G083300*) were upregulated at 96 hai in the susceptible genotype, aligning with Padder et al. (2016), and their activity may facilitate fungal survival, since sugar metabolism is critical for pathogen maintenance in necrotrophic lesions [25]. Nodulin-like genes, involved in sucrose transport, further enhance pathogen fitness during host colonization [91].

In the set of putative S-genes, beyond the phenylpropanoid pathway activation, additional enrichment was observed for secondary metabolite biosynthesis, including flavonoids, isoflavonoids, alkaloids, and terpenoids. These pathways contribute to phytoalexin production, antimicrobial metabolites synthesized in response to pathogen infection [96], but their effectiveness depends on rapid accumulation at infection sites [97]. A UDP-glucosyl transferase stood out as a central hub in the S-gene network, being induced by multiple stress-related signals, such as deoxynivalenol, salicylic acid, ethylene, and jasmonic acid [98]. Moreover, glutamate biosynthesis was enriched among putative S-genes (S9 Table). During pathogen interactions, glutamate metabolism can lead to either "resistance," preserving cell viability, or "evasion," facilitating cell death. Hemibiotrophic pathogens exploit this metabolic shift to establish and persist within the host [99].

Several resistance loci have been associated with the response of common bean to race 65 of *C. lindemuthianum* [14,20,50–52,100]. In our analysis, DEGs near the *Co*-1, *Co*-4, and *Co*-6 loci on Pv01, Pv08, and Pv11 were identified. Near *Co*-1, a phospholipase C-encoding gene was upregulated in the resistant genotype and integrated into its putative R-gene coexpression network. This enzyme hydrolyzes phospholipids to generate diacylglycerol (DAG), which is phosphorylated into phosphatidic acid, a key second messenger in pathogen-triggered signaling [101]. Phospholipid signaling is well-documented in *Arabidopsis* upon *Pseudomonas syringae* infection [102]. On Pv08, a cluster of kinase genes with malectin domains was identified. One was upregulated at 96 hai in the resistant genotype and linked to its putative R-gene network. These proteins function as cell wall sensors, detecting structural disturbances crucial for plant survival [103]. Similar proteins have been mapped to anthracnose resistance loci in the common bean genome [21]. Near *Co*-6, an upregulated GATA transcription factor gene was present in the resistant genotype's putative R-gene network. The interaction between GATA transcription factors and the jasmonic acid pathway has been reported in the *Brachypodium distachyon – Magnaporthe oryzae* pathosystem [82].

The resistance of *Ouro Vermelho* to *C. lindemuthianum* race 65 (Lv134) has been reported as monogenic [16], with SNP markers mapped to Pv04. In our study, however, no genes near this region showed significant differential expression at 48 or 96 hai, nor did they appear in coexpression networks, suggesting that earlier or later stages of infection, or post-transcriptional regulation, may underlie their role. This can be attributed to the dynamic nature of gene regulation. It is possible that the critical transcriptional changes occurred at earlier or later time points than those sampled (48 and 96 hai), a common limitation in time-course studies. Alternatively, the function of the resistance gene in this region may be regulated at the post-transcriptional level, such as through protein modification, without a corresponding change in mRNA abundance.

In contrast, several genes on Pv02, including an alpha/beta-hydrolase, a PLATZ transcription factor, and a hydroxyproline-rich glycoprotein, were strongly expressed and integrated into resistant coexpression networks. This is consistent with previous mapping of the *Co-u* and *CoPv02cX* loci to Pv02 in *BAT 93* and *Xana*, respectively, both associated with resistance to race 65 [11,12,54]. These findings highlight that while Pv04 remains genetically linked to resistance, Pv02 harbors transcriptionally active candidates that may play a central role in the defense response. Although resistance to race 65 (isolate Lv134) in Ouro Vermelho is under monogenic control (16) at genetic level, the extensive transcriptional reprogramming we observed in this study reveals that this single resistance gene might acts as a master switch, activating a broad cascade of defense pathways required to establish the robust resistance phenotype.

Given the race-specific nature *C. lindemuthianum* resistance in common bean, this study offers novel and comprehensive perspective on the molecular response of resistant and susceptible genotypes to race 65. Our findings identify a broad network of genes and pathways involved in resistance, offering valuable targets for the common bean breeding community for the development of future *C. lindemuthianum*-resistant cultivars.

## Supporting information

**S1 Fig. Expression profiles of genes differentially expressed by the cultivars Ouro Vermelho and Estilo, at 0, 48, and 96 hours after inoculation.**
(TIF)

**S1 Table. Genes, primers, and probes for qPCR assay.**
(XLSX)

**S2 Table. Number of RNA sequencing reads for samples of the cultivars BRS Estilo and Ouro Vermelho at different hours after inoculation (hai) with the isolate Lv 134 of race 65 *Colletotrichum lindemuthianum* (0 hai, 48 hai, and 98 hai).** The genomic reference used for sequencing alignment was *Phaseolus vulgaris* v2.1.
(XLSX)

**S3 Table. Differential expression analysis results for the different comparisons performed.**
(XLSX)

**S4 Table. R48-like genes, S48-like genes, R96-like genes, and S96-like genes.**
(XLSX)

**S5 Table. Putative S genes at 48 hpi.**
(XLSX)

**S6 Table. Putative R genes at 96 hpi.**
(XLSX)

**S7 Table. Putative S genes at 96 hpi.**
(XLSX)

**S8 Table. R48-like genes, S48-like genes, R96-like genes, and S96-like genes.**
(XLSX)

**S9 Table. Gene Ontology (GO) enrichment terms of differentially expressed putative R and putative S genes in the different contrasts considering the times 0, 48, and 96 after inoculation of the Ouro Vermelho (resistant) and BRS Estilo (susceptible) common bean cultivars with the Lv 134 isolate of race 65 of *Colletotrichum lindemuthianum*.**
(XLSX)

**S10 Table. Metabolic pathways associated with the putative resistance genes and putative susceptibility genes identified from different contrasts involving the BRS Estilo and Ouro Vermelho cultivars, inoculated with the Lv 134 isolate of race 65 *Colletotrichum lindemuthianum*, at 0, 48, and 96 hours after inoculation.**
(XLSX)

**S11 Table. Degree centrality measures for the gene coexpression network modeled for the Ouro Vermelho cultivar, highlighting the putative resistance genes.**
(XLSX)

**S12 Table. Degree centrality measures for the gene coexpression network modeled for the BRS Estilo cultivar, highlighting the putative resistance genes.**
(XLSX)

**S13 Table. Degree centrality measures for the gene coexpression network modeled for the Ouro Vermelho cultivar, highlighting the putative susceptibility genes.**
(XLSX)

**S14 Table. Degree centrality measures for the gene coexpression network modeled for the BRS Estilo cultivar, highlighting the putative susceptibility genes.**
(XLSX)

## Acknowledgments

We thank the Laboratory of Disease Resistance, Department of Biology, Federal University of Lavras, for providing the infrastructure for inoculation experiments, as well as the genotype samples and fungal isolate.

## Author contributions

**Conceptualization:** Caroline Marcela da Silva Dambroz, Welison Andrade Pereira.

**Data curation:** Caroline Marcela da Silva Dambroz, Alexandre Hild Aono, Larissa Carvalho Costa, Evandro Novaes, Welison Andrade Pereira.

**Formal analysis:** Caroline Marcela da Silva Dambroz, Alexandre Hild Aono, Larissa Carvalho Costa, Evandro Novaes, Welison Andrade Pereira.

**Funding acquisition:** Welison Andrade Pereira.

**Investigation:** Caroline Marcela da Silva Dambroz, Alexandre Hild Aono, Larissa Carvalho Costa, Evandro Novaes, Welison Andrade Pereira.

**Methodology:** Caroline Marcela da Silva Dambroz, Alexandre Hild Aono, Larissa Carvalho Costa, Evandro Novaes, Welison Andrade Pereira.

**Project administration:** Welison Andrade Pereira.

**Resources:** Welison Andrade Pereira.

**Software:** Alexandre Hild Aono, Welison Andrade Pereira.

**Supervision:** Welison Andrade Pereira.

**Validation:** Caroline Marcela da Silva Dambroz, Alexandre Hild Aono, Larissa Carvalho Costa, Evandro Novaes, Welison Andrade Pereira.

**Visualization:** Alexandre Hild Aono, Larissa Carvalho Costa, Welison Andrade Pereira.

**Writing – original draft:** Caroline Marcela da Silva Dambroz, Alexandre Hild Aono, Larissa Carvalho Costa, Evandro Novaes, Welison Andrade Pereira.

**Writing – review & editing:** Caroline Marcela da Silva Dambroz, Alexandre Hild Aono, Larissa Carvalho Costa, Evandro Novaes, Welison Andrade Pereira.

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
