## [Decision Letter · Decision Letter 0]

18 Dec 2024

PONE-D-24-50843Comparative transcriptome analysis provides new insights into the response of common bean to infection by race 65 of Colletotrichum lindemuthianumPLOS ONE

Dear Dr. Pereira,

Thank you for submitting your manuscript to PLOS ONE. After careful consideration, we feel that it has merit but does not fully meet PLOS ONE’s publication criteria as it currently stands. Therefore, we invite you to submit a revised version of the manuscript that addresses the points raised during the review process.

**ACADEMIC EDITOR:  **  I suggest the authors carefully follow the reviewers comments and make the necessary corrections to the manuscript. 

We look forward to receiving your revised manuscript.

Kind regards,

Karthikeyan Adhimoolam

Academic Editor

PLOS ONE

Journal Requirements:

Reviewers' comments:

Reviewer's Responses to Questions

**Comments to the Author**

1. Is the manuscript technically sound, and do the data support the conclusions?

Reviewer #1: Yes

Reviewer #2: Yes

2. Has the statistical analysis been performed appropriately and rigorously? 

Reviewer #1: Yes

Reviewer #2: Yes

3. Have the authors made all data underlying the findings in their manuscript fully available?

Reviewer #1: No

Reviewer #2: Yes

4. Is the manuscript presented in an intelligible fashion and written in standard English?

Reviewer #1: Yes

Reviewer #2: No

5. Review Comments to the Author

Reviewer #1: Bean anthracnose race 65 and its biotypes are widespread in Brazil. Although the introduction is fine. It would have been better to tell a reader how important race 65 and its biotypes are for Brazil.

L116 to L118. Have the dominant alleles Ouro Vermelho known? If yes please describe those alleles. A few recessive alleles also impart bean anthracnose resistance like co-6 and co-Indb. So, remodify the statement

There is a mention of susceptibility genes in GO analysis. Susceptibility genes are those which can be edited to make plants resistant. Instead of using the word susceptibility please remodify it to the appropriate word. This may confuse readers

I was unable to find Figs mentioned in the text. So, I can’t comment on those. Authors should provide Figs with the manuscript.

Because of these reasons, I am unable to move ahead with the manuscript. I am attaching a pdf that I reviewed. You will see that this pdf is without Figs and main Tables

Minor Comments

L92. More studies are on the interaction of beans with race 73. Please check literature

L93 (2021) should be replaced with the correct number [--]

L94 (2016) should be replaced with the correct number [--]

Reviewer #2: The manuscript has merit but is too long and needs modification in material and methods, results and discussion. It is necessary became short this topics. Many information should be in Supplemental information.

6. PLOS authors have the option to publish the peer review history of their article (what does this mean? ). If published, this will include your full peer review and any attached files.

**Do you want your identity to be public for this peer review?** For information about this choice, including consent withdrawal, please see our Privacy Policy .

Reviewer #1: **Yes: ** Bilal A Padder

Reviewer #2: No

---

## [Author Response · Author response to Decision Letter 1]

14 Feb 2025

Response to Reviewers [PONE-D-24-50843]

We sincerely appreciate the reviewers’ valuable contributions, which have helped improve the quality of our manuscript. Below, we provide detailed responses to the comments.

3. Have the authors made all data underlying the findings in their manuscript fully available?

Reviewer #1: No

Response: We apologize for the inconvenience. All figures have now been incorporated and included at the end of the PDF. Additionally, the raw transcriptome data are being deposited in the NCBI database (BioProject ID: PRJNA1219552), ensuring full transparency and availability.

4. Is the manuscript presented in an intelligible fashion and written in standard English?

Reviewer #2: No

Response: The manuscript has undergone a thorough English revision to enhance clarity and precision.

5. Review Comments to the Author

Reviewer #1:

1) Bean anthracnose race 65 and its biotypes are widespread in Brazil. Although the introduction is fine. It would have been better to tell a reader how important race 65 and its biotypes are for Brazil.

Response: We have added information to the introduction to better contextualize its relevance. (L59-72)

2) L116 to L118. Have the dominant alleles Ouro Vermelho known? If yes please describe those alleles. A few recessive alleles also impart bean anthracnose resistance like co-6 and co-Indb. So, remodify the statement.

Response: We have revised the section to include the requested information on dominant and recessive alleles associated with anthracnose resistance. (L51-54)

3) There is a mention of susceptibility genes in GO analysis. Susceptibility genes are those which can be edited to make plants resistant. Instead of using the word susceptibility please remodify it to the appropriate word. This may confuse readers.

Response: We appreciate the observation. We have revised the entire text to ensure terminological accuracy and avoid potential ambiguities. We refer to the differentially expressed genes as ‘putative resistance genes’ (putative R48-genes or putative R96-genes), ‘putative susceptibility genes’ (putative S48-genes or putative S96-genes), ‘resistance-like genes’ (R48-like genes; R96-like genes), and ‘susceptibility-like genes’ (S48-like genes; S96-like genes), as described in the section "Differential expression analysis and enrichment of Gene Ontology terms" (L163-170). We believe these genes can only be definitively classified as resistance or susceptibility after further studies confirm their specific functions.

4) I was unable to find Figs mentioned in the text. So, I can’t comment on those. Authors should provide Figs with the manuscript. Because of these reasons, I am unable to move ahead with the manuscript. I am attaching a pdf that I reviewed. You will see that this pdf is without Figs and main Tables

Response: We have incorporated all figures and tables.

Minor Comments

L92. More studies are on the interaction of beans with race 73. Please check literature

Response: We reviewed the literature and added relevant information. (L89-101)

L93 (2021) should be replaced with the correct number [--] (L96)

Response: References have been corrected as requested.

L94 (2016) should be replaced with the correct number [--] (L95; L93)

Response: References have been corrected as requested.

Reviewer #2: The manuscript has merit but is too long and needs modification in material and methods, results and discussion. It is necessary became short this topics. Many information should be in Supplemental information.

Response: We appreciate the feedback. We conducted a thorough and in-depth review of the entire text, from the Abstract to the Discussion, to make it more concise while ensuring that no content was compromised. The original version contained 8,367 words (including Table 1 and the legends for Figures 1-6). After the adjustments, the manuscript was reduced to 5,985 words, representing an approximate 30% reduction. We believe we have achieved a more direct and objective main text.

Final Consideration:

We thank the reviewers once again for their time spent reviewing our manuscript and for their suggestions. We hope the revisions and improvements meet their expectations. We are available for any further clarifications and look forward to the next stage of the evaluation process.

Best regards

---

## [Decision Letter · Decision Letter 1]

6 Aug 2025

PONE-D-24-50843R1Comparative transcriptome analysis provides new insights into the response of common bean to infection by race 65 of Colletotrichum lindemuthianumPLOS ONE

Dear Dr. Pereira,

Thank you for submitting your manuscript to PLOS ONE. After careful consideration, we feel that it has merit but does not fully meet PLOS ONE’s publication criteria as it currently stands. Therefore, we invite you to submit a revised version of the manuscript that addresses the points raised during the review process.

**ACADEMIC EDITOR: ** I suggest the authors carefully follow the reviewers comments and make the necessary corrections to the manuscript. ==============================

We look forward to receiving your revised manuscript.

Kind regards,

Karthikeyan Adhimoolam

Academic Editor

PLOS ONE

Journal Requirements:

Reviewers' comments:

Reviewer's Responses to Questions

**Comments to the Author**

1. If the authors have adequately addressed your comments raised in a previous round of review and you feel that this manuscript is now acceptable for publication, you may indicate that here to bypass the “Comments to the Author” section, enter your conflict of interest statement in the “Confidential to Editor” section, and submit your "Accept" recommendation.

Reviewer #1: All comments have been addressed

2. Is the manuscript technically sound, and do the data support the conclusions?

Reviewer #1: Yes

3. Has the statistical analysis been performed appropriately and rigorously? 

Reviewer #1: Yes

4. Have the authors made all data underlying the findings in their manuscript fully available?

Reviewer #1: Yes

5. Is the manuscript presented in an intelligible fashion and written in standard English?

Reviewer #1: Yes

6. Review Comments to the Author

Reviewer #1: Thank you for providing the figures along with the manuscript. Although you have addressed all comments in the revision, inconsistencies still exist. For example, Phaseolus vulgaris is often written in full and sometimes as P. vulgaris. The same applies to Colletotrichum lindemuthianum. Authors can write their full names once and then refer to them as P. vulgaris and C. lindemuthianum.

The gene models listed in the text must be italicized

L354. The genes must be italicized

L373. [59] please add author “xyz [59]”

L376. The pathogen is biotrophic for 72 hours and then switches to necrotrophic thereafter. Please refer to Mahiya-Farooq (2019) and Padder et al. (2016) for clarification.

Most of the discussion revolves around the repetition of results. The authors are encouraged to provide a more thorough discussion of these results.

L451, L470, L534. Italicize the words Arabidopsis and Seclerotinia

Authors must check the references. All scientific names in the title should be italicized. Please double-check each cited reference.

7. PLOS authors have the option to publish the peer review history of their article (what does this mean? ). If published, this will include your full peer review and any attached files.

**Do you want your identity to be public for this peer review?** For information about this choice, including consent withdrawal, please see our Privacy Policy .

Reviewer #1: No

---

## [Author Response · Author response to Decision Letter 2]

2 Sep 2025

Round 2

Response to Reviewers [PONE-D-24-50843]

Dear Reviewers,

We sincerely thank you for the insightful comments and constructive suggestions, which have substantially improved the clarity and scientific rigor of our manuscript. We have carefully addressed each point, and the corresponding revisions are highlighted in the tracked version of the manuscript for ease of review.

Reviewer Comment 1: Phaseolus vulgaris is often written in full and sometimes as P. vulgaris. The same applies to Colletotrichum lindemuthianum. Authors can write their full names once and then refer to them as P. vulgaris and C. lindemuthianum.

Response: We appreciate this important point regarding scientific nomenclature. We have standardized the usage of species names throughout the manuscript. The full scientific names Phaseolus vulgaris and Colletotrichum lindemuthianum were introduced upon their first mention in the Abstract and Introduction. Subsequently, their abbreviated forms (P. vulgaris and C. lindemuthianum) were consistently used throughout the remainder of the manuscript, except in section titles, figure captions, and table titles

Reviewer Comment 2: The gene models listed in the text must be italicized.

Response: We thank the reviewer for this correction. We have carefully reviewed the entire manuscript and italicized all gene models as per scientific convention.

Reviewer Comment 3: L354. The genes must be italicized.

Response: Thank you for your correction. We have now corrected this issue and have ensured that all gene names are properly italicized according to standard scientific practice.

Reviewer Comment 4: L373. [59] please add author “xyz [59]”.

Response: This context has been revised in the “Discussion” section to ensure accuracy, and the citation has been corrected accordingly.

Reviewer Comment 5: L376. The pathogen is biotrophic for 72 hours and then switches to necrotrophic thereafter. Please refer to Mahiya-Farooq (2019) and Padder et al. (2016) for clarification.

Response: We thank the reviewer for pointing out this inaccuracy regarding the pathogen's life cycle description. We have revised the statement to correctly reflect the duration of the biotrophic phase, as supported by the references provided.

Changes implemented:

• Lines 383–388 now state: “This is followed by infection vesicle formation and primary hyphae differentiation, marking the biotrophic phase, which lasts for approximately 72 hours, with the host cell remaining alive [60]. The fungus then transitions to the necrotrophic phase, characterized by secondary hyphae and host cell death [25,61,62]. Macroscopic symptoms appear at 96 hai [60,63], which was confirmed in the susceptible cultivar (BRS Estilo) in this study.”

Reference added:

62. Mahiya-Farooq, Padder BA, Bhat NN, Shah MD, Shikari AB, Awale HE, et al. Temporal expression of candidate genes at the Co-1 locus and their interaction with other defense-related genes in common bean. Physiol Mol Plant Pathol. 2019;108: 101424. doi:10.1016/j.pmpp.2019.101424

Reviewer Comment 6: Most of the discussion revolves around the repetition of results. The authors are encouraged to provide a more thorough discussion of these results.

Response: We acknowledge the reviewer's valuable feedback on the discussion section. We have significantly revised this section to provide a more in-depth interpretation and contextualization of our results, moving beyond mere repetition. We focused on elaborating the implications of our findings, connecting them more explicitly to existing literature and broader biological understanding of plant-pathogen interactions.

Reviewer Comment 7: L451, L470, L534. Italicize the words Arabidopsis and Sclerotinia.

Response: We have revised the manuscript to italicize all occurrences of the scientific names Arabidopsis and Sclerotinia, in accordance with nomenclature standards.

Reviewer Comment 8: Authors must check the references. All scientific names in the title should be italicized. Please double-check each cited reference.

Response: We conducted a careful review of all references, using the Mendeley tool to support the revisions. Subsequently, each entry in the list was checked for compliance with the journal’s formatting guidelines. In particular, all scientific names, including taxonomic genera, appearing in the titles of the cited articles have been properly italicized.

On behalf of all co-authors, we sincerely thank the reviewers and the editorial team for their constructive feedback. We believe that the revisions have considerably improved the clarity, accuracy, and scientific value of the manuscript, and we look forward to its reconsideration. We remain at the reviewers’ and editorial team’s disposal for any further clarifications.

Sincerely,

On behalf of all co-authors,

Prof. Welison Andrade Pereira

Federal University of Lavras (UFLA), Brazil

---

## [Decision Letter · Decision Letter 2]

1 Oct 2025

Comparative transcriptome analysis provides new insights into the response of common bean to infection by race 65 of Colletotrichum lindemuthianum

PONE-D-24-50843R2

Dear Dr. Pereira

We’re pleased to inform you that your manuscript has been judged scientifically suitable for publication and will be formally accepted for publication once it meets all outstanding technical requirements.

Kind regards,

Karthikeyan Adhimoolam

Academic Editor

PLOS ONE

Additional Editor Comments (optional):

Reviewers' comments:

Reviewer's Responses to Questions

**Comments to the Author**

1. If the authors have adequately addressed your comments raised in a previous round of review and you feel that this manuscript is now acceptable for publication, you may indicate that here to bypass the “Comments to the Author” section, enter your conflict of interest statement in the “Confidential to Editor” section, and submit your "Accept" recommendation.

Reviewer #1: All comments have been addressed

2. Is the manuscript technically sound, and do the data support the conclusions?

Reviewer #1: Yes

3. Has the statistical analysis been performed appropriately and rigorously? 

Reviewer #1: Yes

4. Have the authors made all data underlying the findings in their manuscript fully available?

Reviewer #1: Yes

5. Is the manuscript presented in an intelligible fashion and written in standard English?

Reviewer #1: Yes

6. Review Comments to the Author

Reviewer #1: The authors have addressed all the comments. It is now suitable for publication in the journal, plos one.

7. PLOS authors have the option to publish the peer review history of their article (what does this mean? ). If published, this will include your full peer review and any attached files.

**Do you want your identity to be public for this peer review?** For information about this choice, including consent withdrawal, please see our Privacy Policy .

Reviewer #1: **Yes: ** Bilal Ahmad Padder

---

## [Editor Report · Acceptance letter]

PONE-D-24-50843R2

PLOS ONE

Dear Dr. Pereira,

I'm pleased to inform you that your manuscript has been deemed suitable for publication in PLOS ONE. Congratulations! Your manuscript is now being handed over to our production team.

Kind regards,

on behalf of

Dr. Karthikeyan Adhimoolam

Academic Editor

PLOS ONE